# Coordinating DNA Replication and Mitosis through Ubiquitin/SUMO and CDK1

**DOI:** 10.3390/ijms22168796

**Published:** 2021-08-16

**Authors:** Antonio Galarreta, Pablo Valledor, Oscar Fernandez-Capetillo, Emilio Lecona

**Affiliations:** 1Genomic Instability Group, Spanish National Cancer Research Centre (CNIO), 28029 Madrid, Spain; a.galarreta@hotmail.com (A.G.); pvalledor@cnio.es (P.V.); 2Science for Life Laboratory, Division of Genome Biology, Department of Medical Biochemistry and Biophysics, Karolinska Institute, S-171 21 Stockholm, Sweden; 3Centre for Molecular Biology Severo Ochoa (CBMSO, CSIC-UAM), Chromatin, Cancer and the Ubiquitin System Lab, Department of Genome Dynamics and Function, 28049 Madrid, Spain

**Keywords:** DNA replication, mitosis, ubiquitin, SUMO, USP7, CDK1

## Abstract

Post-translational modification of the DNA replication machinery by ubiquitin and SUMO plays key roles in the faithful duplication of the genetic information. Among other functions, ubiquitination and SUMOylation serve as signals for the extraction of factors from chromatin by the AAA ATPase VCP. In addition to the regulation of DNA replication initiation and elongation, we now know that ubiquitination mediates the disassembly of the replisome after DNA replication termination, a process that is essential to preserve genomic stability. Here, we review the recent evidence showing how active DNA replication restricts replisome ubiquitination to prevent the premature disassembly of the DNA replication machinery. Ubiquitination also mediates the removal of the replisome to allow DNA repair. Further, we discuss the interplay between ubiquitin-mediated replisome disassembly and the activation of CDK1 that is required to set up the transition from the S phase to mitosis. We propose the existence of a ubiquitin–CDK1 relay, where the disassembly of terminated replisomes increases CDK1 activity that, in turn, favors the ubiquitination and disassembly of more replisomes. This model has important implications for the mechanism of action of cancer therapies that induce the untimely activation of CDK1, thereby triggering premature replisome disassembly and DNA damage.

## 1. The Ubiquitin and SUMO System

Post-translational modifications (PTMs) play key roles in the cell as essential regulators of the spatiotemporal control of protein function. Among the PTMs, ubiquitin (Ub) is a small protein that is conjugated to lysine residues on target proteins and mark them for degradation by the proteasome, a process that was discovered 40 years ago [1]. Ubiquitination is a three-step process, where ubiquitin is first conjugated to the E1 activating enzyme, then transferred to an E2 conjugating enzyme and finally attached to a specific substrate through the cooperation of ubiquitin E3 ligases and their partner, E2. The system achieves great specificity thanks to the existence of more than 600 E3 ligases in the human genome. SUMO (small ubiquitin-like modifier) is a 100 amino acid ubiquitin-like protein that was discovered in the 1990s and displays high homology with ubiquitin [2]. The SUMOylation pathway also involves a single E1 enzyme that transfers SUMO to UBC9, the only E2 enzyme [3,4]. UBC9 transfers SUMO to its targets by itself or in association with a limited number of specific E3 SUMO ligases [5].

The ubiquitin system can generate a slew of different configurations from a single ubiquitin conjugated to one or several lysine residues (monoubiquitination and multi-monoubiquitination) or ubiquitin chains, either linear or branched. In mammals, SUMO2/3 also forms chains and can also participate in mixed ubiquitin/SUMO chains. This “ubiquitin code” establishes a set of specific modifications with different functional outcomes by modifying the stability, localization, activity or interactome of the target proteins. One of the main transducers of ubiquitin and SUMO is the AAA ATPase VCP (valosin containing protein), also known as p97 segregase, a central factor in the regulation of protein homeostasis [6,7,8,9] that mobilizes the chromatin-bound substrates conjugated to ubiquitin [10,11,12] or ubiquitin-like molecules [13,14,15,16]. To this end, VCP uses a repertoire of adaptors/cofactors which recognize poly-ubiquitinated substrates through their ubiquitin-binding domains [17,18]. These cofactors associate with VCP in a hierarchical system, where the primary cofactors determine the binding of additional adaptors to determine specific VCP–adaptor configurations that recognize the different substrates [19,20].

Both ubiquitination and SUMOylation can be reverted by the action of specific proteases called deubiquitinases (DUBs) [21] and SUMO-specific proteases [22]. Finally, there is crosstalk between ubiquitination and SUMOylation with a specialized set of E3 ubiquitin ligases, known as SUMO-targeted ubiquitin ligases (STUbLs), which recognize and ubiquitinate SUMOylated proteins [23]. Conversely, some DUBs have been shown to be active on SUMO and SUMOylated proteins, including USP7 [24] and USP11 [25]. Ub and SUMO constitute a complex post-translational modification system that affects many essential cellular functions, including the regulation of DNA metabolism and genome stability [26].

## 2. SUMO and Ubiquitin in DNA Replication 

DNA replication mediates the accurate copy of the entire genetic material of a cell. Thus, it needs to be tightly regulated to preserve genome integrity and prevent the accumulation of DNA damage and replicating errors that lead to cancer development and the onset of aging. DNA replication can be divided into three phases: initiation, the S phase or elongation and DNA replication termination. The ubiquitin/SUMO pathways are essential regulators of DNA replication in each of these phases (Table 1). 

DNA replication initiation is a two-step process involving origin licensing in G1 and the subsequent formation of the pre-replication complex (pre-RC) that is required to transition into the S phase. Origin licensing takes place in all potential replication origins by the sequential loading of two protein complexes. First, the origin recognition complex (ORC1-6) binds origins of replication. Next, the mini-chromosome maintenance (MCM) complex is recruited with the help of the ATPase CDC6 and CDT1 (CDC10-dependent transcript 1) to form the pre-RC [76]. However, only a subset of these origins is activated or fired in each S phase [77]. A recent report has identified OBI1 as an E3 ubiquitin ligase that multi-monoubiquitinates ORC3 and ORC5 to promote the firing of a subset of origins of replication without affecting the assembly of the pre-RC (Figure 1) [27]. To avoid re-replication of the DNA in the S phase, the origins need to be fired only once per cell cycle, and the ubiquitin system also prevents the re-licensing of replication origins after they have been fired. Both ORC1 and CDT1 are poly-ubiquitinated after the initiation of DNA replication by the SCF^Skp2^ ubiquitin–ligase complex (Figure 1). Consequently, they are extracted from chromatin by VCP and undergo proteasome-mediated degradation [28,29].

Once the pre-RC is assembled, origin activation marks the onset of the S phase by triggering the unwinding of the DNA double helix to allow loading of the replication machinery. Origin activation requires the phosphorylation of several subunits of the MCM complex by the Ser/Thr protein kinases DBF4-dependent kinase (DDK) and by interphase cyclin dependent kinases (CDKs) [78,79]. This phosphorylation promotes the assembly of the replicative CMG helicase through recruitment of the GINS complex and CDC45 [80]. The phosphorylation of the MCM complex is controlled by ubiquitin/SUMO in two steps (Figure 1). First, SUMOylation of MCM proteins limits their phosphorylation in G1, preventing premature origin activation [66]. In the transition to the S phase SUMOylation levels decline as the phosphorylation of the MCM complex rises. Second, the ubiquitin/SUMO pathways also switch off the origin activation signal by timing the degradation of the DDK. Chromatin-bound DDK is SUMOylated, leading to its ubiquitination by STUbLs to induce its degradation by the proteasome. The SUMO protease Ulp2 protects DDKs from degradation, allowing the early steps of DNA replication to occur [63].

After the origins have been activated, the double MCM2-7 hexamer divides and establishes two replication forks through the recruitment of additional replication factors including RPA, RFC, PCNA and the replicative DNA polymerases [77]. Proteomic analyses of chromatin under replication have revealed that DNA replication forks are embedded in a SUMO-rich environment [81,82,83]. Although the exact functions of replisome SUMOylation remain to be elucidated, a recent report showed that the SUMOylation of the catalytic subunit of DNA polymerase ε is important for DNA replication in yeast [64,65]. In addition to the SUMOylation of specific factors, we have proposed that the collective SUMOylation of the replication machinery supports DNA replication by creating an environment that facilitates interactions among replication factors [84], analogous to the SUMO-based group modification model proposed by Stefan Jentsch for DNA repair [85]. Within this model, we previously showed that USP7 is a SUMO-dependent deubiquitinase that maintains low levels of ubiquitination in the replisome, and whose action is necessary to sustain DNA replication [24]. This is particularly relevant in light of the recent advances showing how ubiquitin and SUMO are essential for disassembly of the replication machinery after DNA replication termination, as we develop in the next section.

## 3. DNA Replication Termination

The study of DNA replication termination is not technically easy and, as a consequence, its molecular mechanisms have remained relatively unexplored compared with the initiation and elongation phases [86,87,88]. Recent studies have shed light on the sequence of events that sets up the end of DNA replication when the two forks converge [89,90,91]. During fork convergence, the action of topoisomerases to release topological stress is impeded by the lack of space ahead of the forks [92]. Instead, this stress is relieved by the clockwise rotation of the two forks, which generates intertwining between the two replicated sister chromatids. This intertwining is finally resolved by Type II topoisomerases or by the action of Pif1 and Rrm3 helicases [93,94,95,96,97,98,99,100,101]. Thanks to these enzymes, the replisome encounter does not induce fork stalling, and the CMG helicases keep moving at the same speed to rapidly pass each other, moving from the leading to the lagging strand of the converging fork [102]. Next, the single-stranded gap between the 3ʹ end of the leading strand and the downstream Okazaki fragment of the opposing fork is filled in and the CMGs travel on dsDNA no longer supporting DNA synthesis but allowing Okazaki fragment processing by Polδ and FEN1 [102]. At this stage, the replication machinery is disassembled from dsDNA in a process that involves the ubiquitination of the CMG helicase and the action of the AAA ATPase VCP [102]. Finally, the sister chromatids need to be decatenated by Type II topoisomerases before chromosome segregation [103,104]. Evidence accumulated in recent years has shown that the disassembly of the replication machinery is one of the key steps in DNA replication termination and that the ubiquitin pathway lies at the heart of this process.

## 4. Replisome Disassembly at the End of DNA Replication

As CMGs cannot be reloaded during the elongation phase, removal of the replisome irreversibly blocks fork progression [105]. Thus, the eviction of the replication machinery is tightly regulated and restricted to replication forks that have reached the downstream Okazaki fragment to ensure full replication of the DNA [102]. The molecular events that control replisome disassembly are only beginning to be understood. One of the events associated with DNA replication termination is the poly-ubiquitination of the MCM7 subunit of the CMG helicase at lysine-48 (K48) that is recognized by VCP through the ubiquitin fusion degradation protein 1 homolog (UFD1L) and nuclear protein localization protein 4 homolog (NPLOC4) heterodimer [70,106,107]. It has been proposed that MCM7 ubiquitination helps in the extraction of many components of the replication machinery from chromatin (Figure 2A) [68,69,71,108]. Notably, while much of the attention has been placed on MCM7 ubiquitination on replisome disassembly during DNA replication, it seems unlikely that this is the only critical ubiquitination event that is involved in DNA replication termination. Many additional replisome components might be similarly ubiquitinated and extracted from chromatin following the same or equivalent pathways, and MCM7 ubiquitination may serve as a model to understand the mechanisms governing ubiquitin-mediated replisome eviction.

There are two basic mechanisms that control ubiquitination-driven replisome disassembly to avoid untimely eviction of the replication machinery: the first one is the spatio-temporal restriction of the action of E3 ubiquitin-ligases; the second one is the ubiquitin threshold for the action of VCP. The E3 ligase in charge of ubiquitinating MCM7 differs between yeast and higher eukaryotes. While yeast SCF (Skp, Cullin, F-box-containing complex) associates with Dia2 to induce MCM7 ubiquitination [68], in higher eukaryotes, the Cullin RING ligase 2 (CUL2) protein binds to LRR1 to modify MCM7 [69,70,71]. How is the activity of these E3 ligases restricted during DNA replication elongation? Recent work has shown that the presence of the lagging strand in active DNA replication forks inhibits the action of the E3 ligases or blocks their access to the replisome. Once the final parental duplex has unwound, the CMGs helicases move on to dsDNA and the interaction with the lagging strand is lost, allowing CMG ubiquitination [109,110]. In yeast, Dia2 interacts with Ctf4 and Mrc1 and it has been proposed that SCF^Dia2^ travels together with the replication machinery [111,112,113,114]. The interaction of the lagging strand template with the CMG helicase would mask the substrates of SCF^Dia2^, although a direct action preventing its association with the replisome has not been completely ruled out [110]. In the case of CUL2^LRR1^, the lagging strand directly prevents the interaction of its LRR domain with the CMG helicase, and thus it is specifically targeted to terminated CMGs [70,71]. The active DNA replication model could explain how a single helicase is disassembled at the end of a telomere or at a single-stranded nick, but it does not justify how CMG ubiquitination is prevented during DNA replication initiation before the replication forks have formed [115].

A second layer of control is established through the length of the ubiquitin chains on the CMG, since the action of VCP depends on a “ubiquitin threshold”, whereby VCP/Ufd1/Npl4 requires four to five ubiquitin molecules to act on its substrates [116]. In yeast, SCF^Dia2^ only generates short ubiquitin chains on MCM7 during DNA replication, most likely due to the inhibition exerted by the interaction of the lagging strand with the CMG [110]. A recent report showed that TIMELESS-TIPIN stimulates the activity of CUL2^LRR1^ in *C. elegans* to achieve efficient ubiquitination of the CMG [117]. The ubiquitination of the CMG could also be limited by specific DUBs, and we have previously identified USP7 as a SUMO-specific deubiquitinase that maintains a SUMO-rich/ubiquitin-poor environment at the DNA replication forks [24,81]. In this sense, SUMO could act as a signal to drive the ubiquitination of replication factors upon DNA replication termination or compete for ubiquitination to prevent the untimely modification of these proteins [84]. In yeast, the MCM2-7 complex has been shown to be SUMOylated in G1, and MCM7 SUMOylation persists during the S phase [66]. Interestingly, proteomic analyses revealed that USP7 inhibition induces the ubiquitination of many replication factors, including MCM7, suggesting it may play a role in preventing the premature disassembly of the replisome [24]. On the other hand, USP7 was previously found to interact and cooperate with the MCM-binding protein (MCM-BP) in unloading MCM7 [118]. Thus, USP7 may have opposing actions during the disassembly of the replication machinery, and how these actions are coordinated has not been clarified yet.

The relevance of ubiquitin-mediated replisome disassembly was further substantiated by *in vitro* experiments in yeast. The minimal requirements for replisome disassembly involve the ubiquitination of the replication machinery, including MCM7, by SCF^Dia2^ and its extraction by Cdc48 in combination with Ufd1/Npl4 [114]. Whether other components of the replication machinery are ubiquitinated and contribute to the recruitment of VCP and the eviction of specific proteins has not been explored yet. In addition, the interplay between MCM7 ubiquitination and SUMOylation remains to be studied. Once MCM7 has been ubiquitinated, little is known about its fate. Although VCP often targets proteins for their proteasomal degradation [7,119], the inhibition of the proteasome has no effect on MCM7 accumulation [120]. These data suggest that the CMG complex is recycled alongside other components of the replication machinery. It would be interesting to determine the fate of the replication factors upon replisome disassembly.

## 5. Replisome Disassembly after DNA Damage and outside the S Phase

Reinforcing the central role of ubiquitination for replisome disassembly, there are two situations outside the canonical pathway after DNA replication termination where MCM7 is also modified and contributes to the removal of the replication machinery from chromatin. First, if the canonical replisome extraction pathway fails, the replication machinery needs to be mobilized before mitosis. In contrast to yeast, work in *C. elegans* and mouse embryonic stem cells has revealed the existence of a back-up mechanism for replisome unloading in mitosis by an alternative pathway independent of CUL2^LRR1^ [69,70,73,74]. Mitotic replisome disassembly involves the TRAIP-dependent ubiquitination of MCM7 at K6 and K63 (Figure 2A) [69,73,74]. After its ubiquitination, MCM7 is extracted by VCP/Ufd1/Npl4 in cooperation with an additional cofactor, UBX domain-containing 3 (UBXN3; the worm ortholog of human FAS-associated factor 1 (FAF1)). The process requires the action of the SUMO protease ULP4 (the worm ortholog of SENP6/7), although its exact functions remain unknown [70] and SUMOylation does not affect this pathway in *Xenopus* egg extracts [74]. Similar to SCF^Dia2^ in yeast, the E3 ligase TRAIP is constitutively associated with the replisome, where it plays a role during the repair of DNA–protein crosslinks [70,71,113,121]. The mechanisms controlling TRAIP activation in mitosis are still unknown. Abrogating both interphase and mitotic disassembly pathways by combined knockdown of LRR1 and UBXN3 stabilizes the presence of CMG on chromatin through the metaphase and leads to synthetical lethality [70]. 

Second, there are several types of DNA damage that require replisome disassembly to allow for repair during the S phase. Common fragile sites (CFSs) and other difficult-to-replicate regions are copied late in the S phase and often induce fork stalling, leading to replication stress and subsequently to an increase in anaphase ultrafine bridges, copy number variations or chromosomal rearrangements [122,123,124,125]. A recent work has linked the appearance of genomic alterations at CFS to premature disassembly of the replisome at these sites. This disassembly is associated with the ubiquitination of MCM7 by TRAIP (Figure 2B). After the eviction of the replication machinery, these structures are repaired by microhomology-mediated end joining (MMEJ), leading to the appearance of chromosomal alterations [73] similar to the ones observed in CFSs. Interestingly, the action of TRAIP at stalled forks is stimulated by CDK1, linking the activation of the mitotic program to the disassembly of the replisome. 

Another instance that induces replisome extraction from chromatin is the presence of single-strand breaks (SSB) (Figure 2C). These breaks are often generated by problems during the action of topoisomerase I, and they are also produced as intermediates in base excision repair (BER) [126]. The repair mechanisms induced upon SSB-induced fork collapse are strand-specific but always involve the removal of CMG from chromatin [72]. When the break is present in the leading strand, the advance of the CMG helicase generates a single-ended double-strand break (seDSB), where the CMG passively slides off (Figure 2C). In contrast, when the break is present in the lagging strand, the fork generates a ssDSB with a single strand overhang while the CMG moves on the dsDNA. Following the same pathway that acts after DNA replication termination, the replication machinery is then ubiquitinated and evicted by VCP (Figure 2C) [69,70,71,109,110]. Last, DNA interstrand crosslinks (ICLs) covalently link both strands of the DNA molecule and block DNA replication and transcription [127]. There are two pathways for ICL repair mediated by the Fanconi anemia (FA) family of proteins and the NEIL3 glycosylase [128,129,130]. Both pathways involve the convergence of the two DNA replication forks and their stalling next to the lesion [131,132], followed by the ubiquitination of MCM7 at K48 by the E3 ligase TRAIP (Figure 2D) [75]. The length of the ubiquitin chains on MCM7 determines the repair pathway that is activated. Longer chains induce CMG unloading by the VCP segregase in both converging forks [75,120,133]. Despite the similarity to DNA replication termination, there is an additional contribution of BRCA1 in promoting CMG ubiquitination and unloading in response to ICL [120]. The removal of the replication machinery is necessary to allow the action of the FA machinery to repair the damage. On the other hand, short ubiquitin chains deposited on MCM7 at K48 by TRAIP activate the NEIL3 glycosylase pathway [75]. NEIL3 directly binds ubiquitinated MCM7 and mediates the unhooking of the ICL that does not require the disassembly of the replication machinery [134,135].

Thus, cells have evolved alternative pathways for replisome disassembly that work during mitosis or in the presence of DNA damage. All these pathways share the ubiquitination of the replication machinery as a central event, but they are mediated by specific mechanisms that involve, in many cases, the E3 ubiquitin ligase TRAIP. How TRAIP is specifically activated upon damage or during mitosis independently of the convergence of DNA replication forks and how its action is blocked during DNA replication termination [68,69,70,71] is still not understood. 

## 6. The Ubiquitin Connection between DNA Replication and CDK1 Activation

The traditional view of the cell cycle considered DNA replication completion and the activation of the mitotic program as independent events separated by the G2 phase. However, there is increasing evidence to support a direct functional link between the end of the S phase and the onset of mitosis. First, the mitotic machinery is activated right after the PCNA foci disappear [136] and the active DNA replication forks suppress mitotic activity [137]. Based on these observations, the Lindqvist lab proposed a model for progression from the S phase to mitosis, where active DNA replication forks work as molecular breaks that prevent premature mitotic entry [138]. This model also implies the existence of a signal during DNA replication termination, when the active DNA replication forks disappear, that deactivates these breaks to start the mitotic program. Interestingly, CMG ubiquitination and replisome disassembly are also suppressed by active DNA replication forks, suggesting that this modification might link DNA replication termination and the progression into mitosis [109,110]. Conversely, the activation of CDK1 promotes the ubiquitination of the CMG and disassembly of the replisome [73]. Together, these observations support the existence of a ubiquitin–CDK1 relay, where the disassembly of terminated replisomes fosters CDK1 activation (Figure 3A). In turn, CDK1 activity reinforces disassembly of the replication machinery from terminated replisomes. Thus, replisome disassembly would not be favored in the early S phase when CDK1 activity is low. The increased CDK1 activity in the late S phase could drive the premature removal of the replication machinery from stalled forks at CFS and difficult-to-replicate regions. 

Complementary to the CDK1-induced ubiquitination of the replisome, we have recently shown that the SUMO/ubiquitin equilibrium at active DNA replication forks also controls CDK1 activation. In 2016, we identified USP7 as a SUMO-dependent deubiquitinase that prevents excessive ubiquitination and SUMOylation in DNA replication forks [24,84]. Of note, the inhibition of USP7 induces the ubiquitination of many replication factors, including MCM7, at the canonical site responsible for replisome disassembly [24]. In parallel to the control of the ubiquitination of the replisome, USP7 also restricts the activity of CDK1 through the regulation of its phosphatase, PP2A [139]. As a consequence, blocking USP7 reduces PP2A activity and leads to the activation of CDK1. Concomitantly, an increase in the ubiquitination of the replisome results in premature disassembly of the replication machinery and generation of CDK1-dependent DNA damage in the S phase. Thus, the ubiquitin system constitutes an additional layer of regulation during cell cycle progression that connects the termination of DNA replication and the activation of CDK1 through regulated disassembly of the replication machinery. Beyond the control imposed by the accumulation of CDK activity, the ubiquitin–CDK1 relay could explain, at least in part, how active DNA replication prevents premature activation of the mitotic program. At the same time, this model provides a mechanism of how the unloading of the replication machinery is stimulated during the late S phase and mitosis to avoid problems after cell division. Whether the group SUMOylation of the replisome also plays a role in setting the stage for the ubiquitination of the CMG remains to be determined. 

In addition to helping us understand how cells coordinate the completion of DNA replication with activation of the mitotic program, the ubiquitin–CDK1 connection has important clinical implications, since many inhibitors in this pathway are currently being studied as anticancer agents. We propose that the activation of CDK1 is an attractive therapeutic avenue, since it will induce DNA damage during DNA replication in combination with premature entry into mitosis that can lead to mitotic catastrophe and cell death (Figure 3B). A similar effect has been observed with ATR inhibitors that combine the generation of damage with the loss of the G2/M checkpoint [140,141]. Working through a different mechanism, the inhibition of USP7 can also elicit CDK1-dependent DNA damage and premature activation of the mitotic program. In this sense, we have shown that CDK1 is essential for the toxic effects of USP7 inhibitors in cancer cells. In addition, mutations that influence the status of CDK1 in cancer cells will be determined for the application of USP7 inhibitors in the clinic. 

## Figures and Tables

**Figure 1 ijms-22-08796-f001:**
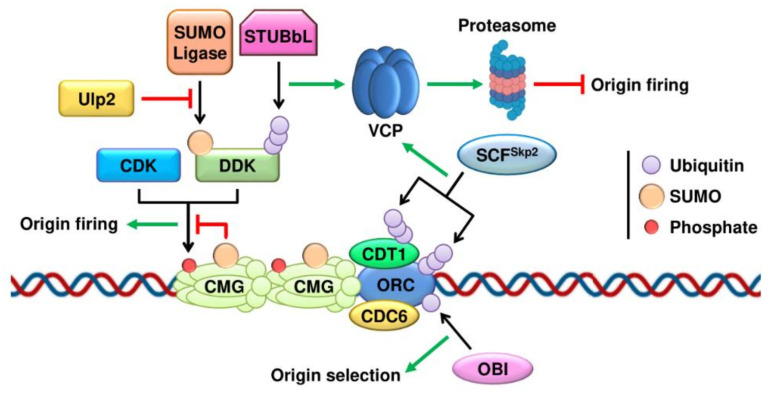
Control of DNA replication initiation by the ubiquitin and SUMO pathways. Model for the assembly of the pre-RC complex and the different layers of control by ubiquitination and SUMOylation of the initiation machinery. CMG is the replicative helicase composed of CDC45, the MCM2-7 complex and the GINS complex.

**Figure 2 ijms-22-08796-f002:**
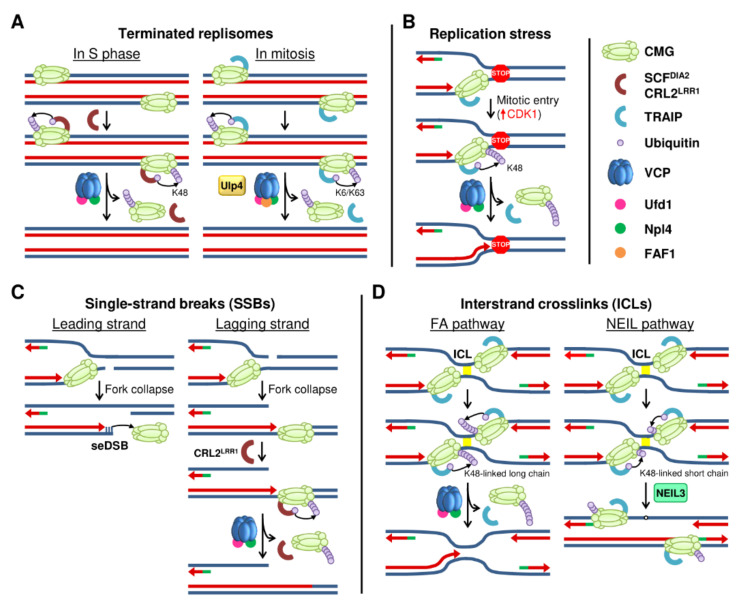
Models for replisome disassembly. (**A**) Disassembly of terminated replisomes during the S phase (left) and in mitosis (right) is mediated by the ubiquitination of MCM7 with specific E3 ubiquitin ligases. (**B**) Replisome disassembly under replication stress is mediated by the CDK1-dependent activation of TRAIP to ubiquitinate MCM7. (**C**) Replisome disassembly in the presence of SSBs. When the SSB is in the leading strand (left), the CMG slides off the break, while a SSB in the lagging strand leads to CMG translocation along dsDNA and its ubiquitination by CRL2^LRR1^ (right). (**D**) In the presence of ICLs, replication forks convergence and MCM7 is ubiquitinated at K48 by TRAIP. Long ubiquitin chains lead to CMG unloading (left) and repair through the FA pathway, whereas short ubiquitin chains activate the NEIL3 glycosylase pathway (right).

**Figure 3 ijms-22-08796-f003:**
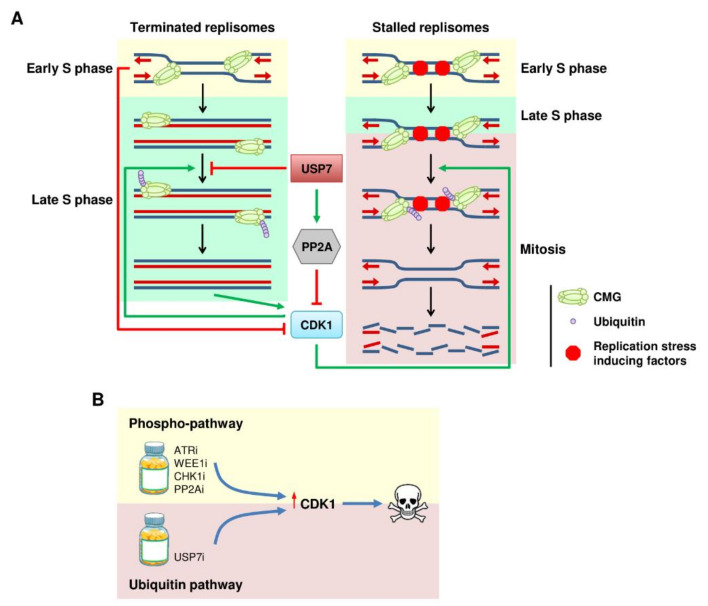
Ubiquitination links DNA replication and CDK1. (**A**) Model for the ubiquitin/CDK1 relay in DNA replication. Active DNA replication forks and USP7 suppress CMG ubiquitination and CDK1 activation to prevent premature entry in mitosis. Accumulating CDK1 activity promotes CMG ubiquitination, leading to disassembly of the terminated replisomes before cell division, or disassembly of the stalled forks inducing DNA damage. (**B**) Therapeutic implications for the ubiquitin–CDK1 connection. Targeting the ubiquitin pathway might be an alternative way to prematurely activate CDK1 and lead to cell death.

**Table 1 ijms-22-08796-t001:** Regulation of DNA replication by Ub and SUMO.

Target	Modification	E3 Ligase	Protease	Organism	Role	Cellular Context	References
ORC3/ORC5	Ubiquitin	OBI1		Human	Origin selection for firing	Replication initiation, G1/S	[27]
ORC1	Ubiquitin	SCF^Skp2^		Human	Prevent re-licensing	Replication initiation, G1/S	[28]
CDT1	Ubiquitin	SCF^Skp2^	USP37	Human	Prevent re-licensing	Replication initiation, G1/S	[29,30,31]
		APC/C-CDH1		Human, *Xenopus*	Regulate licensing. Prevent licensing in quiescence.	M/G1	[32,33]
		CRL4-DDB1^CDT2^		Human, yeast, *Xenopus*, *C. elegans*, zebrafish	Prevent re-licensing. Prevent licensing upon DNA damage	Replication initiation, G1/S	[31,34,35,36,37,38,39,40,41]
CDC6 (Cdc18 in yeast)	Ubiquitin	APC/C-CDH1		Human	Prevent re-licensing.	Early G1	[42]
					Prevent pre-RC assembly	Quiescence	[43]
		SCF^CyclinF^			Prevent re-licensing	G2	[44]
		SCF^Cdc4^		Yeast		G1/S	[45,46,47]
Geminin	Ubiquitin	APC/C-CDH1	Dub3, USP7	*Xenopus*	Promote origin licensing	M/G1	[48,49,50]
Treslin	Ubiquitin	CRLs		Human	Prevent origin firing	G1/S	[51]
CDC45	Ubiquitin	APC/C-CDH1		Human	Prevent origin firing	M/G1	[52]
Claspin	Ubiquitin	APC/C-CDH1	USP28, USP29, USP9X	Human	Prevent origin firing. Recovery from G2 checkpoint response.	M/G1	[53,54,55]
		SCF^βTrCP^	USP7	Human	Recovery from checkpoint response	G2/M	[56,57,58,59]
DBF4	Ubiquitin	APC/C-Cdc20		Yeast	Prevent re-replication. Prevent new pre-RC firing	M/G1	[60,61,62]
DDK	SUMO	Siz1, Siz2	Ulp2	Yeast	Prevent origin firing	Replication initiation	[63]
	Ubiquitin	Slx5/Slx8		Yeast			[63]
Polymerase ε (Pol2 subunit)	SUMO	Smc5/6 complex (Mms21 subunit)		Yeast	Promote fork progression under replication stress	Elongation, S phase	[64,65]
MCM7	SUMO	Mms21, Siz1, Siz2	Ulp2	Yeast	Prevent origin firing	Replication initiation, G1	[66,67]
	Ubiquitin	SCF^Dia2^		Yeast	Trigger replisome disassembly	Replication termination	[68]
		CRL2^LRR1^		*C. elegans*, *Xenopus*, mouse embryonic stem cells			[69,70,71]
				*Xenopus*		Under lagging strand SSBs	[72]
		TRAIP		*C. elegans*, *Xenopus*, mouse embryonic stem cells	Trigger replisome disassembly	Mitosis	[69,73,74]
				*C. elegans*, *Xenopus*	Trigger replisome disassembly	Stalled replisomes upon RS	[73]
				*Xenopus*	Trigger replisome disassembly	ICL repair by FA pathway	[75]
					NEIL3 recruitment	ICL repair by NEIL3 glycosylase pathway	[75]

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
