# Peer review of "Coordinating DNA Replication and Mitosis through Ubiquitin/SUMO and CDK1"

_ijms, 2021, doi:10.3390/ijms22168796_

Round 1

Reviewer 1 Report

Galarreta et al.  Comments for the authors

This is a nice and comprehensive review of how the ubiquitin system regulates DNA replication, focusing on replisome disassembly, and its interplay with CDK1 activation and mitotic entry.  Overall the manuscript is very well structured and written, and only few minor points should be addressed before acceptance for publication.

  • Sections on replisome disassembly models (and figure 2) partially overlap with a recent review by Priego Moreno and colleagues (BIochem Soc Trans 2020, PMID: 32490508). However, here the authors specially focus on the regulation of those pathways by ubiquitin and SUMO modifications. Therefore, as a suggestion, I would include a table summarizing the main ubiquitin/SUMO targets and their E3 ligases and role, if known, organism, etc along different phases and cellular contexts of replication, such as, ORC3/ORC5-OBI1 in initiation, MCM-7-TRAIP for elongation, etc.

  • Last section on the proposed ubiquitin-CDK1 relay is very attractive. According to the text and the published data, CDK1 activation would lead to premature replisome disassembly and entry into mitosis with uncomplete DNA repair and DNA damage. However, in the model depicted in figure 3A, it seems to me cells are entering mitosis with assembled replisomes, suggesting a delay in disassembly rather than a premature eviction form chromatin. Moreover, from the figure it is not suggested the presence of DNA damage or incomplete replication. I would try to redraw the figure to better fit the proposed model.

  • There are some minor differences in nomenclature between figures and text, and few missing details, which I would modify/correct to facilitate going through the manuscript, especially for non-specialized readers:

Lane 51-52: VCP, also known as p97 segregase

                        Lane 76: S phase or elongation

                        Lane 99: origin firing or activation

Lane 102-103: I would refer as interphase CDKs to separate from CDK1

Section 2: Clarify CMG helicase as composed of MCM complex + GINS complex and CDC45 (maybe add this to figure legend on Figure 1).

Lane 273: refer to fork stalling as a source of replication stress to match with Figure 2B

Lane 334-335: cite reference for modification of TRAIP by CDK1 (Deng et al.  2019)

Figure 3: Symbol legend similar to Figure 2 is missing.

Author Response

We want to thank the reviewer for his/her kind comments. As suggested we have included a Table indicting the main E3 ligases and deubiquitinases involved in the control of DNA replication. We have also updated Figure 3A to better reflect the timing of replisome eviction and the presence of damage. In addition, we have corrected all the details indicated by the reviewer to make the manuscript more easy to read. We hope you find the changes address your concerns.

Reviewer 2 Report

In their manuscript, Galarreta, Valledor and colleagues present a literature review of the current knowledge regarding the implication of ubiquitin and CDK1 in DNA replication and mitosis. The 2 first sections of the manuscript introduce the ubiquitin and SUMO systems and their relation to DNA replication; the following sections of the review focus on DNA replication, replisome disassembly and the transition from S phase to mitosis; the sixth – and last – section of the manuscript highlight the ubiquitin connection between DNA replication and CDK1 activation.

The review is well written and informative.

Minor comments:

  1. In their abstract, the authors conclude about the important implication of their model for cancer therapies. This aspect is solely mentioned in the very last paragraph of the manuscript. Maybe the authors should consider adding this aspect/discussion to a short conclusion or to a new section at the end of the review (depending on how much they want to comment or focus on this aspect).
  2. The keywords are missing.
  3. Both ubiquitin and SUMO are very commented on across the manuscript. However, the title does not mention SUMO. The authors might consider updating their title if they deem it necessary.
  4. The figures appear quite pixelized. If a higher resolution is available for the figures, those would be preferable.

Author Response

We are happy that the reviewer finds the manuscript well written and informative. We understand that adding a section on the implication of this work on cancer therapies would be interesting. However we feel that it would complicate the message in the review as the full analysis of the implications of this interplay in cancer therapy would require to make an extensive and detailed analysis that is beyond our aim. Following the reviewer's suggestions we have included the keywords, updated the title and improved the quality of the figures. We thank the reviewer for these comments.

Reviewer 3 Report

In this manuscript, Galarreta et al. have reviewed the roles of ubiquitin and SUMO in the coordinated regulation of replication and mitosis. The manuscript is well written, provides a good overview of the current state of knowledge and discusses possible areas of future investigation. While the role of ubiquitin and SUMO in regulating DNA replication has been extensively reviewed elsewhere, discussions on the role of these modifiers in replication termination, and linking these modifications to CDK1 activation, does provide uniqueness to the article.

I believe the manuscript is acceptable in the current state. Although there are some typographical errors, these will be easily addressable during copy-editing.

Author Response

We would like to thank the reviewer for his/her appreciation of our work.